# Knowledge, Attitudes, and Practices Related to Antibiotic Use and Antibiotic Resistance among Poultry Farmers in Urban and Peri-Urban Areas of Ouagadougou, Burkina Faso

**DOI:** 10.3390/antibiotics12010133

**Published:** 2023-01-10

**Authors:** Abdallah Sawadogo, Assèta Kagambèga, Arshnee Moodley, Abdoul Aziz Ouedraogo, Nicolas Barro, Michel Dione

**Affiliations:** 1Laboratory of Molecular Biology, Epidemiology and Surveillance of Foodborne Bacteria and Viruses, Department of Biochemistry-Microbiology, Doctoral School of Science and Technology, Joseph KI-ZERBO University, Ouagadougou 03 BP 7021, Burkina Faso; 2International Livestock Research Institute, Nairobi 00100, Kenya; 3Department of Veterinary and Animal Sciences, University of Copenhagen, 1817 Frederiksberg, Denmark; 4National Institute of Statistics and Demography, Ouagadougou 01 BP 374, Burkina Faso; 5International Livestock Research Institute, Dakar BP 24265, Senegal

**Keywords:** knowledge, attitudes, practices, antimicrobial resistance, chicken, west Africa

## Abstract

Increased use of antibiotics in livestock is a public health concern, as it poses risks of antibiotic residues and antibiotic-resistant pathogens entering the food chains and infecting humans. A cross-sectional survey was conducted on 216 poultry farms to study knowledge, attitudes and practices of poultry farmers on the use of antibiotics in urban and peri-urban areas of Ouagadougou. Results show that only 17.13% (37/216) of farmers attended training on poultry production. Majority of farmers—85.65% (185/216) were not knowledgeable about the rational use of antibiotics. When there was a disease outbreak, 31.98% (63/197) of farmers used veterinary drugs without a prescription and 22.34% (44/197) consulted a community animal health worker. It should also be noted that 79.19% (156/197) of farmers reported using chicken meat as per normal if the bird died during or right after treatment with an antibiotic. Knowledge of rational use of antibiotics was positively influenced by a good attitude adopted by the farmer during the illness of birds and negatively influenced by disease treatment success and high level of education of the farmer. Lack of knowledge about the rational use of antibiotics including their use without a prescription are serious risk factors for the emergence of antimicrobial resistance. Awareness of farmers and other veterinary drug supply chain actors such as drug stockists and animal health workers on best practices in antimicrobial use and promotion of good biosecurity on farms are important to reduce the misuse of antibiotics.

## 1. Introduction

Intensification of livestock production has led to increased use of veterinary drugs, including antibiotics [1]. Antibiotics are used for therapy, prophylaxis, meta-phylaxis, and growth promotion [2]. Antimicrobial resistance (AMR) is one of ten global health challenges [3,4]. Studies have indicated that AMR of livestock origin is increasing in low- and middle-income countries (LMICs), with the largest increase in poultry and pigs [5]. From 2000 to 2018, the proportion of antimicrobial compounds with resistance higher than 50% (P50) increased from 0.15 to 0.41 in chickens [5]. Emergence of antimicrobial resistance in microorganisms is a natural phenomenon, yet antimicrobial resistance selection has been driven by antimicrobial exposure in healthcare, agriculture, and the environment and the interactions that exist between these components [6]. Imprudent or irrational use is hypothesized to be more pronounced in low-income countries where veterinary services are limited, poor animal husbandry is practiced, and access to antimicrobials is poorly controlled [7]. The use of antimicrobials without veterinary guidance and supervision can result in incorrect treatment and/or inappropriate dosing, which when coupled with insufficient observation of withdrawal periods before slaughter, can result in the final meat products containing antibiotic residues and antimicrobial-resistant pathogens [8]. In LMICs, such as Burkina Faso, where small-scale production is still dominant and with a generally low awareness about AMR among the farmers, lack of good legislation and other regulatory mechanisms have been shown to be less successful in reducing non-necessary use of antibiotics [9]. In Burkina Faso, self-medication is practiced by 74.60% of poultry farmers and growth promoters are used in 93.65% of poultry farms [10]. In addition, there is no monitoring and effective control of sales and use of veterinary drugs [11,12,13]. However, the practices, knowledge, and attitudes that motivate livestock farmers to use antibiotics are still poorly understood in most LMIC farming communities. Furthermore, the lack of a national livestock database makes it difficult to track animals, vaccination, disease management, and document treatment at the animal level [7]. In view of these challenges and limitations, a bottom-up approach showing farmers how to reduce the need for antibiotics and use them in a medically rational way can complement existing antimicrobial use (AMU)-reducing initiatives and governance frameworks. However, this bottom-up approach requires a thorough understanding of the drivers for antibiotic use [14], current knowledge levels, and access to veterinary advice and drugs, which will also improve our understanding on how to develop and deploy context-relevant AMU-reducing interventions for urban and peri-urban poultry farmers in Burkina Faso. Therefore, the objective of our study was to evaluate knowledge, attitudes and practices of poultry farmers on the use of veterinary drugs with a focus on antibiotics in urban and peri-urban poultry farmers in Ouagadougou, Burkina Faso.

## 2. Materials and Methods

### 2.1. Ethical Clearance

The study was approved by the ethical committee of the Ministry of Health, Burkina Faso, with reference number 2020-9-186. Informed consent was obtained from each participant before he/she was interviewed. Consequently, all participants gave their consent to participate in the study.

### 2.2. Study Areas and Sampling

A cross-sectional study was conducted between March and July 2020 in urban and peri-urban poultry farms of Ouagadougou. It is the most densely populated city, with 2,415,266 inhabitants [15], and considering the high demand for meat and eggs, poultry farmers settle in and around the city to meet this need. A total of 216 poultry farms (broilers and layers) were selected for the study. From each farm, the manager (the owner or designated worker) was requested to participate in the study.

### 2.3. Sample Size Calculation

The sample size was calculated using the following conditions: in a study in Ghana, expected proportion of 12.1% of farmers who adopted good practices (farmers sought individual prescriptions from the veterinary office before purchasing drugs for the birds) [16], a risk of error α of 5%, and a confidence level of 95%. Based on this estimate, the required sample size was 163 poultry farms. This number was increased to 216 to account for missing data and the size of the previous study.

### 2.4. Data Collection

The questionnaire used in the study was based on the ‘Antimicrobial use in livestock production systems’ (AMUSE Livestock tool) developed by the International Livestock Research Institute (ILRI) [17] to facilitate harmonized data collection regarding farmers’ knowledge, attitudes and practices across the CGIAR research programs within the AMR conceptual framework in LMICs, and modified versions of the questionnaire have been used in Ethiopia [18] and Uganda [14].

The questionnaire consisted of five sections: (1) demographic information, such as the age of the farmers, education level, and sex, (2) chicken farms’ characteristics, (3) classes and frequency of drugs used in farms, (4) knowledge of antibiotics use, and (5) biosecurity practices and attitudes encountered on the farms. The questionnaire was translated into French and interviews were conducted either in French or in the local language (Mooré) depending on the level of education of the farmer. Responses were recorded using a database, Open Data Kit (ODK), a source-based smartphone platform that can be used to create electronic questionnaire forms for real-time data entry.

### 2.5. Data Management and Analysis

Statistical analyses were performed with the STATA/SE 16.0 software. First, a univariate descriptive analysis was carried out to obtain an overall view of the data. This made it possible to define the characteristics of the farms surveyed, the most used practices, and the level of knowledge of antibiotics. For classes with less than five respondents, groupings were made to guarantee the reliability of the test, and in the case of the latter, Fisher’s exact test was used.

Finally, to identify the potential determinants of the level of knowledge on antibiotic use, a multivariate logistic regression was used with the variable of interest. The outcome variable (knowledge about the prudent use of antibiotics) was defined as follows:

Antibiotics are used (systematically) to treat sick animals: the answer is yes as antibiotics are mainly used to treat infection.

Antibiotics are used (systematically) to prevent disease in animals: The answer is wrong as antibiotics are not systematically used to prevent disease. However, they can be used as a prophylactic in some circumstances.

Antibiotics are used (systematically) to fatten animals: The answer is wrong as antibiotics are not systematically used to fatten animals. However, they can be used as a growth promoter.

We coded 1 if the individual has a good knowledge of prudent use of antibiotics (answered yes to all three questions) and 0 otherwise (fail at least one out of three question). Bivariable analyses using *χ*^2^ tests were performed. Twenty-one variables with a *p*-value ≤ 0.05 during bivariable analysis were included in the multivariate analysis. Unconditional logistic regression with stepwise backward elimination was used to obtain the final model, which retained 11 variables. These variables mainly explain the level of knowledge about prudent use of antibiotics. The statistical significance levels used in all estimations were 1%, 5%, and 10%.

## 3. Results

### 3.1. Sociodemographic Characteristics of Poultry Farmers

The sociodemographic characteristics of the respondents are presented in Table 1. Of the 216 respondents, 199 (92.13%) were men while only 17 were women (7.87%). Ages ranged between 17 and 52 years, with >50% of respondents being between 17 and 32 years. Most farmers had either primary or secondary level education. Sixty-two respondents (28.71%) had no formal education and only thirty-seven farmers (17.13%) said they attended a training on poultry production.

### 3.2. Characteristics of Poultry Farms

The main farm characteristics are shown in Table 2. Majority of farms kept >500 birds, with broiler farms being predominant (66.67%), and majority (93.98%) practicing total confinement. Most farmers (77.31%) used grain or crop residues, and 96.76% of farmers used commercial chicken feed pre-mixes. Chicken feces were used by 89.35% as fertilizer in their own farms/gardens or elsewhere. In addition to poultry, 27.31% and 22.68% of the farmers owned small ruminants (sheep or/and goats) and cattle, respectively, on the same premises as the chicken farms.

### 3.3. Veterinary Drugs’ Use and Vaccines

Types of veterinary drugs and vaccines used in the last four weeks prior to the survey are shown in Figure 1. The most reported drugs used on the farms were vitamins/iron supplements (46.74%) and antibiotics (43.46%). The least used were acaricides (0.69%).

Figure 2 describes which classes of antibiotics were used in the past month at least once. Out of 100 antibiotics shown, tetracyclines were the most common (23.02%), followed by macrolides with 11.06%. The least used group of antibiotics were fluoroquinolones (0.23%).

Respondents reported using tetracyclines four times or more in the month (*n* = 95, 43.98%). Similarly, 59 respondents described using macrolides 3 times or more in the month, however 133 (61.86%) reported not using macrolides at all in the last 4 weeks. As for the vitamin/iron supplements, 182 (84.25%) respondents used them at least once a week.

### 3.4. Knowledge of Antibiotic Use by Farmers

The assessment of knowledge on prudent use of antibiotics was conducted by asking a series of three questions (Figure 3). Good knowledge was deemed as answering correctly to all three questions. When asked if antibiotics are (systematically) used to treat sick animals, 93.98% answered correctly, “yes”. When asked whether antibiotics are used (systematically) to prevent disease on the farm, 85.65% got this wrong. When asked whether antibiotics are used (systematically) to fatten animals quickly, 58.33% got it wrong. The correct answer to each of the three previous questions, i.e., answering the first question in the affirmative and the second and third questions in the negative, is considered good knowledge. According to our categorization, only 11 respondents (5.09%) had good knowledge of prudent use of antibiotics.

### 3.5. Biosecurity Practices and Attitudes towards Prudent Use of Antibiotics

Biosecurity practices and attitudes observed during the study are presented in Table 3. More than half (53.70%) of the respondents reported use of veterinary medicine to keep their chickens healthy. Regarding the last disease occurrence, 114 (52.78%) reported to have had an episode 1–6 months ago. The main type of infections were respiratory diseases (53.57%), followed by digestive/intestinal diseases (36.73%). When an infection appears on the farm, 31.98% of respondents noted use of veterinary drugs without a prescription. Between 19% and 23% of respondents consulted a veterinarian (private or public) or a community animal health worker. Less than 30% reported using a licensed veterinarian in the last two months. Veterinary drugs were largely bought from a veterinary pharmacy or agrovet shop (55.84%) and majority of respondents (94.85%) reported administering the last medication to the birds by themselves. Seventy-nine percent (79%) of respondents reported normal (sale or consumption) of the eggs or chicken meat during or shortly after treatment with an antibiotic and as well as the sale of dead birds.

## 4. Factors That Influence Practices and Attitudes

Farm size was associated to practices such as: attitude when the animals were sick (*p* = 0.005), referral to a professional for diagnosis and treatment (*p* = 0.025), and drug acquisition channels (*p* = 0.000).

Education level of the farmer was associated to practices such as: referral to a professional for diagnosis and treatment (*p* = 0.032) and drug acquisition channels (*p* = 0.002). The results of the statistical tests are presented in Table 4.

Multivariate analysis demonstrated that the last occurrence of a disease on the farm and good attitude adopted during the illness significantly and positively influenced the degree of knowledge of prudent use of antibiotics at the 5% threshold. Treatment success, referral to the right health professional for medication advice, and a high level of education had a significant, negative influence on the degree of knowledge of prudent use of antibiotic use at the 5% threshold (Table 5). The complete table of Logistic regression results for the multivariate model is presented in the Appendix A (Table A1).

## 5. Discussion

Good knowledge and good practices have been widely reported as predictors of correct KAP regarding AMU and AMR [19,20]. In our study, half of the respondents reported the systematic use of antibiotics to prevent diseases and one third reported using antibiotics without a prescription. Only 19–23% of respondents sought advice from either an animal health community worker or a veterinarian. This rate is lower than that reported by Njoga et al. [21]. To reduce imprudent antibiotic use, consultation with a qualified animal health professional is necessary to make a correct diagnosis and advise on the appropriate therapy; moreover, their practices are governed by regulations [22]. Key interventions could include improving farmers’ access to affordable veterinary services and providing information about appropriate use of antibiotics.

Almost 90% of respondents noted the use of manure as a fertilizer for vegetable gardens of farm crops. Only a few respondents noted leaving the fecal droppings on the farm. Poultry manure is used as an organic fertilizer in the agriculture field globally, which contains antibiotics that create several environmental and human health-related issues. These antibiotics persist in the environment for a long period of time and ultimately reached the human body through contaminated crops and vegetables [23]. Dumping untreated manure, leaving it in the open, or using untreated manure as a fertilizer may result in the spread of residues, AMR, and pathogens into the environment or into food crops that are subsequently consumed by humans or other animals [24]. In addition to poultry, farms had other livestock species on the same site. The presence of multiple animal species on the same farm could constitute a risk not only for the selection and transmission of pathogens but also for AMR between animals [25].

Seventy percent of respondents noted utilizing eggs and chicken meat from birds undergoing treatment, recently treated, or that had died. This constitutes poor biosecurity practice and increases the risk of the spread of infectious diseases among the human population.

Amongst the most used medicines were vitamins/iron supplements and antibiotics (43–47%). It is unclear if these supplements contain sub-therapeutic concentrations of antibiotics. Nonetheless, this frequent use of vitamins suggest some poultry production challenges, and issues such as heat stress can make the use of vitamins inefficient [26].

Respondents understood the role of antibiotics for therapy, prophylaxis, and growth promotion, which is reflected in their frequent antibiotic use, as shown in Figure 2, i.e., 23.02% of respondents described using tetracycline and 11.06% using macrolide at least once in the four weeks preceding the interview. However the prudent use of antibiotics was an issue. While 56% of respondents obtained their veterinary drugs from a veterinary pharmacy or agrovet, 32% reported purchasing antibiotics without a prescription. While more than half the respondents procured their medicines from a presumably licensed pharmacy/agrovet, antibiotic use practices were still poor, highlighting the complexity of the antibiotic use situation since the sale of antibiotics is a business. It was reported by Dione et al. [27] in a study in Uganda that antibiotics were the most cost-effective class of drugs.

For future interventions to improve medically rational antibiotic use in livestock to be effective and sustainable, not only should farmers be targeted but also veterinary drug shops and animal health service providers. This approach has been used to reduce the misuse of over-the-counter human medicines, thus preventing the sale of medicines without a prescription from a doctor [28].

The size of the farm was a factor that positively influenced certain practices. Those who tended to consult a qualified veterinarian in case of disease outbreaks were mainly those with larger farms. In addition, the larger the size of the farm, the more likely it was that those in charge would pay for veterinary drugs in formal drug stores rather than in informal markets. Education level was also a factor that positively influenced some practices. The higher the education level of the respondents, the more likely they were to use a qualified veterinarian for diagnosis and treatment of poultry. The higher the level of education, the more likely respondents were to pay for veterinary drugs from official pharmacies.

The last occurrence of an infection and the attitude adopted during the disease episode had a significant and positive influence on the degree of knowledge of antibiotic use. Indeed, the better the individual’s attitude during the illness, the more likely it is that his or her level of knowledge about the prudent use of antibiotics is higher. At the onset of illness, the prudent approach would be to contact animal health professionals, which would reinforce the good knowledge by following their advice and prescriptions. The more the actors use good practices, the more it positively influences their knowledge on the prudent use of antibiotics. In the case of antibiotics, the technical support provided by the prescribing veterinarian influences their use [29]. A study in the Netherlands found that veterinarians with favorable attitudes towards the prudent use of antimicrobials were positively affecting their farmer clients and, as a result, reducing AMR [30].

The success of treatment had a significant, negative influence on the degree of knowledge of antibiotic use. Contrary to what was expected, the more effective the treatment administered by the individual, the more likely they were to have poor knowledge of antibiotic use. If the treatment was successful, it could maintain their belief that antibiotics could be used to effectively prevent disease. In practice, the farmer may consider antibiotics as a factor of production and decide to use them according to the expected gain from the treatment [31]. However, this widespread use could pose a danger later by promoting the emergence of antimicrobial-resistant microorganisms.

The higher the level of education, the more the respondents used antibiotics for prevention or as growth promoters, all of which is misused. This could be explained by the fact that the higher the level of education, the more means (middle class) farmers have to buy antibiotics, which leads to their widespread use for prevention or as growth promoters. Moreover, contrary to what was expected, even those who often used a qualified veterinarian used antibiotics imprudently. This could be explained by the self-medication practiced by respondents. In fact, in the event of illness, after the veterinarian’s visit, the drugs prescribed were subsequently used beyond the veterinarian’s indications. The farmer seeks to secure his/her income by avoiding a major epidemic on the farm. This situation has been described in a survey conducted in the rabbit farming sector, showing a negative correlation between the income level and antibiotic consumption [32]. This was shown in Table 5, which indicated the practice of self-medication. Indeed, the more successful the treatment administered by the individual, the more likely he/she is to have poor knowledge of antibiotic use, keeping them in bad practices. According to Om and McLaws [33], widespread use of antibiotics occurred on all farms and was driven by four factors: the belief that antibiotics are necessary for animal husbandry, limited knowledge on the risks, easy access to antibiotics, and weak monitoring and control systems. Low-income, small-scale, semi-intensive farmers are focused on the benefits of food animal production as it is linked to improved livelihood, and little attention is paid to the consequences of antibiotic use as farmers may not be directly impacted by it. High farm-level usage of antimicrobials was correlated with high levels of resistance of Salmonella in poultry in Nigeria [34]. There is no compilation of data on microbial resistance on a national scale, but different studies have been carried out showing multidrug-resistant bacteria isolated from farm animals, especially poultry [11,35].

Our study shows low levels of knowledge of farmers about prudent use of antibiotics. There is a need to increase awareness amongst farmers on the prudent use of antibiotics in food animals but there is also an urgent need to provide farmers with support to improve their animal husbandry practices; moreover, better regulation is also needed. It is necessary to reinforce the capacities of the various laboratories in charge of the control and monitoring of AMR [36]. The multisectoral national action plan to combat AMR expired in 2020 and a new plan is being elaborated for Burkina Faso. In this new plan, the legal status should be strengthened by developing specific legislation for the fight against AMR. In addition, the laboratories in the universities carry out sensitivity tests as part of their research, which is an additional factor for the emergence of AMR, hence the need to involve them directly in the fight against AMR. Associate drug companies should have a veterinary and technical sales representative assigned to clients such as livestock farmers to educate and monitor farm operations while selling their products (day-old chicks, poultry equipment, feed additives, etc.). This helps reduce the careless use of antibiotics and provides knowledge to the farmers.

## 6. Conclusions

This study demonstrated the poor practices of poultry farmers on the rational use of antibiotics in Ouagadougou. These poor practices constitute a major problem that hinders the productivity of the poultry farms and constitutes a public health problem because of the interactions that exist between animals and humans, especially in areas of proximity such as urbans zones. Moreover, AMR disproportionately affects LMICs who have a high infectious disease burden, compounded by the relatively easy access to antibiotics and frequent over-the-counter sales without a prescription. There is a need to raise awareness and train all relevant stakeholders on rational use and the animal and human health risks related to misuse of antibiotics. To reduce irrational use of antibiotics in poultry farming in Burkina Faso, our results suggest targeting the following areas: reinforce the awareness of the actors on the whole poultry production chain, strengthen access to quality veterinary services, and set up multi-sectoral platforms to implement and monitor the national AMR strategy.

## Figures and Tables

**Figure 1 antibiotics-12-00133-f001:**
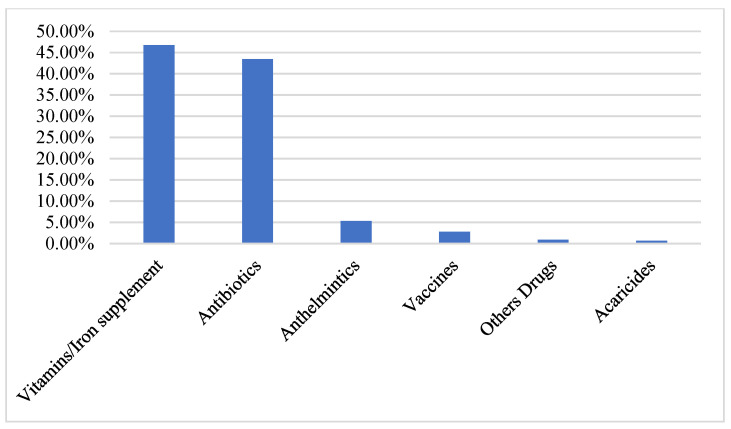
Types of veterinary drugs used in chicken farms.

**Figure 2 antibiotics-12-00133-f002:**
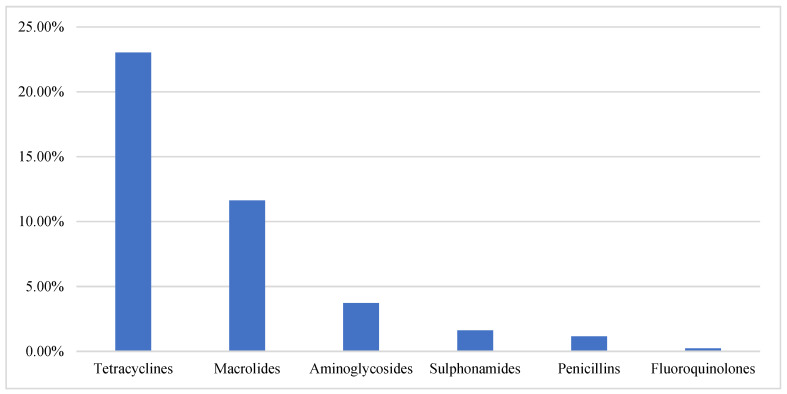
Antibiotic classes used on chicken farms in the past month.

**Figure 3 antibiotics-12-00133-f003:**
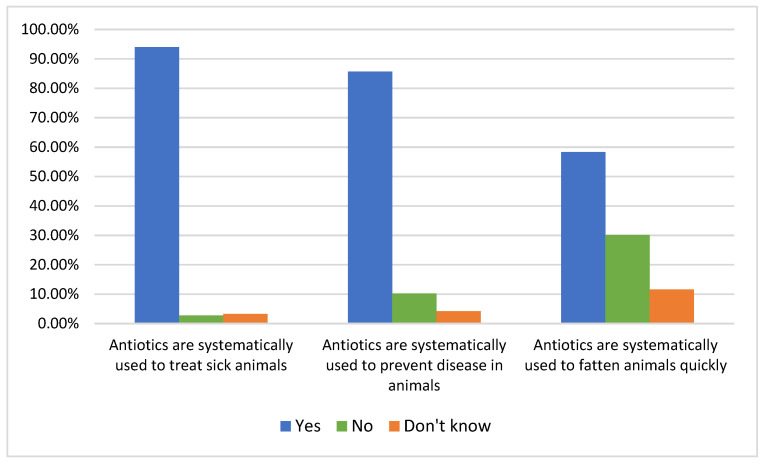
Knowledge of poultry farmers on the prudent use of antibiotics.

**Table 1 antibiotics-12-00133-t001:** Sociodemographic characteristics.

	Category	Effective (Percentage)
Sex	Man Woman	199 (92.13%) 17 (7.87%)
Age	17–32 32–42 ≥42	124 (57.40%) 60 (27.78%) 32 (14.82%)
Education level	No formal education Primary Secondary Academic	62 (28.71%) 79 (36.57%) 66 (30.56%) 7 (3.24%)
Training in poultry production	Yes No	37 (17.13%) 179 (82.87%)

**Table 2 antibiotics-12-00133-t002:** Main characteristics of farms visited.

	Category	Effective (Percentage)
Farm size	140–499 500–999 ≥1000	60 (27.77%) 80 (37.03%) 76 (35.18%)
Type of production	Layers (egg production) Broilers Day-old chick production All the above	54 (25%) 144 (66.67%) 4 (1.85%) 14 (6.48%)
Husbandry management system	Housed day and night (total confinement/intensive) Free by day, housed at night (Semi-intensive) Free day and night (Extensive/free-range)	203 (93.98%) 12 (5.56%) 1 (0.46%)
Use of grain or crop residues as chicken feed	Yes No	167 (77.31%) 49 (22.69%)
Use of household waste to feed the chickens	Yes No	85 (39.35%) 135 (60.65%)
Use of commercial pre-mixes	Yes No	209 (96.76%) 7 (3.24%)
Manure	Used as fertilizer Sold it Gave it away Used as biofuel Dumped on the farm premises Other	193 (89.35%) 9 (4.17%) 9 (4.17%) 2 (0.93%) 2 (0.93%) 1 (0.46%)
Possession of other animals	Sheep/goats Cattle Horse/donkey Pig Rabbit	59 (27.31%) 49 (22.68%) 6 (2.77%) 6 (2.77%) 2 (0.92%)

**Table 3 antibiotics-12-00133-t003:** Biosecurity practices and attitudes encountered on the farms.

	Category	Effective (Percentage)
Precautions to prevent disease	Use of veterinary medicines (including vaccines) Clean/disinfect farm Improve feeding Other	116 (53.70%) 74 (34.26%) 25 (11.57%) 1 (0.46%)
Period of last disease occurrence on the farm	<1 month ago 1–6 months ago 7–12 months ago >12 months ago Never been sick	72 (33.33%) 114 (52.78%) 7(3.27%) 4 (1.85%) 19 (8.80%)
Type of disease	Respiratory disease Digestive/intestinal disease Sudden death Skin diseases/wounds Neurological signs Other	105 (53.57%) 72 (36.73%) 7 (3.57%) 2 (1.02%) 4 (2.04%) 6 (3.06%)
Attitudes adopted in case of disease occurrence	Use of veterinary drugs without a prescription Consult a public veterinarian Consult a community worker Consult a private veterinarian Using traditional medicines Medications applied/left by the veterinarian Other Consult a traditional healer Never been sick	63 (31.98%) 45 (22.84%) 44 (22.34%) 38 (19.29%) 2 (1.02%) 2 (1.02%) 2 (1.02%) 1 (0.51%) 19 (8.80%)
Use of a licensed veterinarian in the last two months	Yes No	59 (27.31%) 157 (72.69%)
Drug acquisition channels	From the veterinary pharmacy/agrovet From an animal service provider From the veto From a market Friends/neighbors Traditional healers From the human pharmacy Street vendors	196 (55.84%) 129 (30.10%) 43 (10.04%) 43 (10.04%) 12 (2.80%) 2 (0.46%) 2 (0.46%) 1 (0.23%)
Person who administered the drug at last use	Myself Veterinarian Employee Community worker Friend	406 (94.85%) 17 (3.97%) 3 (0.93%) 1 (0.23%) 1 (0.23%)
What do you do with the eggs of sick chickens/birds during and sometime after treatment?	Use normally (sell or consume) Mix with other eggs Throw away	156 (79.19%) 3 (1.52%) 38 (19.29%)
What do you usually do if a sick chicken/bird dies sometime after treatment?	Use normally Bury the dead animal Burn the dead animal Throw away	172 (79.63%) 4 (1.85%) 10 (4.63%) 30 (13.89%)

**Table 4 antibiotics-12-00133-t004:** Influence of farm size and education level on certain practices and attitudes (results of the KHI2 test).

Factor	Practice	*p*-Value
Farm size	Attitude when the animals were sick	0.005
	Referral to a professional for diagnosis and treatment	0.025
	Drug acquisition channels	0.000
Education level	Referral to a professional for diagnosis and treatment	0.032
	Drug acquisition channels	0.002

**Table 5 antibiotics-12-00133-t005:** Logistic regression results for the multivariate model.

Variables	Coefficients	*p*-Value
Poultry farm size	−0.465	0.191
Eating the eggs of animals that have just been treated with drugs	0.848	0.116
Drug acquisition channels	0.920	0.122
Use of the help of a professional, such as a licensed veterinarian, in the last 2 months	−1.602	0.075 *
Treatment success	−1.748	0.001 ***
Last time a chicken/bird was sick	0.665	0.003 ***
Use of laboratory services, such as a blood sample test, in the past 12 months	−0.116	0.905
Good practice when your chickens/birds were sick	2.145	0.003 ***
Referral to a professional for advice on medications	−1.549	0.032 **
Education level	−0.619	0.007 ***
Participation in a livestock training course	−1.383	0.123

Significant at thresholds of 1% (***), 5% (**), and 10% (*).

## Data Availability

The data presented in this study are available upon request from the corresponding author. The data is not publicly available because the study contains several other components that will be the subject of a future manuscript.

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
