# Peer review of "Knowledge, Attitudes, and Practices Related to Antibiotic Use and Antibiotic Resistance among Poultry Farmers in Urban and Peri-Urban Areas of Ouagadougou, Burkina Faso"

_antibiotics, 2023, doi:10.3390/antibiotics12010133_

Round 1

Reviewer 1 Report

Knowledge, Attitudes and Practices related to antibiotic use and antibiotic resistance among poultry farmers in urban and peri-urban areas of Ouagadougou, Burkina Faso

The manuscript describes the surveillance results of knowledge and awareness of antibiotics use in poultry farmers in Burkina Faso. Overall, the manuscript is well written and well designed. I only have a few notes and suggestions:

1.      Go through the manuscript to correct any typos and errors, I found some.

2.      Do not use abbreviated terms for keywords!

3.      L65-71: please revise the formatting!

4.      Please provides institutional review boards if any, at the beginning of method section.

5.      How you determine the samples?

6.      How many farmers in total in that area?

7.      L192: please explain what is KAP

8.      L265: a survey a survey?

9.      L269: according to (32) -> please mention the name of the author, for example: according to Om & McLaws (32)…

10.   Is there any difference among the farm sizes with the knowledge, awareness, and practical imprudent use of antibiotics? i.e., smaller farmers would tend to have lower knowledge and awareness. As you have considerably large samples, that would be great to test such different using statistical method, and it would be useful to develop a priority of actions related to the poor practice in using antibiotics.

11.   I finely understand the situation that access to vet is not that easy. Creating and implementing regulations for such problem is also challenging if not supported by sufficient number of qualified veterinary practitioners. I am wondering how the roles of poultry company (feed manufacturers, breeders, vet drugs distributor etc.) to help with educational program to increase the awareness of the farmers? For instance, in my country (I am from developing country too), the companies mostly have veterinary representative, together with technical sales representative assigned to specific area to educate and monitor the farms operation while selling their products (day old chick, poultry equipment, feed additive etc.). That’s effective to reduce the imprudent use of antibiotics and to transfer knowledge to the farmers.

12.   As you mention the AMR, is there an evidence or study about the AMR status from Burkina Faso? If so, that would support your argument. 

Author Response

Please see the attachment response to Reviewer 1  comments.

Reviewer 2 Report

The authors of the manuscript addressed important public issues - AMR  and food safety but some amendments are required as detailed below

Comments for authors

  1. Line 19: put a full stop between “antibiotics” and “More”
  2. Line 20: “When a disease appeared on the farm….” Do you mean “When there is a disease outbreak….”?,
  3. Line 21: Who are “community workers” please? It’s not clear, to the international community,  whom these category of workers are.
  4. Lines 22-24: “Some practices are serious risk factors for the emergence of antimicrobial resistance” List the practices and state the proportions of the respondents involved. Populate the knowledge factors with figures or values (e.g. 42.5%, 94/216).  
  5. Line 34-36: Replace “showing” with “exhibiting”; do you mean 0.15% to 0.41%? I think the figure might be more than that. Consider more recent paper (2021-2022) for an accurate/updated figure/data on the global AMR status in poultry.
  6. Lines 36-37: Apart from imprudent antibiotic usage, environmental, climatic and behavioural factors can also drive AMR in LMICs. For instance migration and overcrowding in refugee camps due to insecurity, flooding due to climate change, animal-human interactions, etc. It will be good to hint on these other causes of AMR (in the context of One Health).
  7. Line 49-51: “The practices, knowledge and attitude that ….”?
  8. Line 54: “veterinary rationale”?
  9. Lines 60-62: to evaluate knowledge, attitude and practices of which category of people? Poultry farmers?
  10. Lines 65-66: Do you mean urban and peri-urban poultry farms in Ouagadougou?
  11. Line 69: How did the authors select the 216 farms surveyed? Which sampling method was used and how was it done?
  12. Line 73-74: What do the authors mean by “epidemiological approach” in sample size calculation? There are free online tools used for sample size estimation in epidemiological studies such as Raosoft® sample size calculator (http://www.raosoft.com/samplesize.htm)
  13. Again, how did the authors arrive at the 15% prevalence used in the sample size calculation? 50% is usually assumed where there is no published data. If the 15% used was published earlier, kindly cite the paper. If not, recalculate the sample size based on 50%.
  14. Data collection: Did the authors’ pilot-test the questionnaire in Burkina Faso before deploying it? The questionnaire may have been successfully used in other countries but country-specific context may vary among countries, necessitating pilot-testing especially in non-English speaking countries.
  15. Table 2: Change “Keeping types” to “Husbandry management system”; Change the categories to “Intensive”, “Semi-intensive” and “Extensive/free-range”
  16. Figure 1: Vaccines are not drugs. I suggest you edit the figure legend/caption to read” Types of veterinary biological uses…”
  17. Lines 161-162: I do not agree with the authors view on respondents’ good knowledge of antibiotic.  Antimicrobials can be used systematically for prophylaxis or growth promotion, depending on the type of antibiotic (based on WHO’ AWaRe classification) and the law guiding antibiotic use in the country. For instance Tetracyclines are still being used for non-therapeutic purposes in livestock production in the US but not in the EU. So, it is country specific. I suspect that the authors may have misinterpreted the responses if the law guiding antibiotic use in Animal agriculture (poultry production) in Burkina Faso was not considered.
  18. Lines 191-193: Compare the Knowledge, attitude and practice towards antibiotic used use in poultry, as found in this study, with that reported for Nigeria (as you have chosen). You did not investigate AMR in Salmonella and hence no basis for the comparison.

The authors may want look at this paper which conducted a similar study like theirs in Nigeria, to compare their findings and cite/reference the paper. Antimicrobial drug usage pattern in poultry farms in Nigeria: implications for food safety, public health and poultry disease management. Veterinaria Italiana, 57 (1): 5-12. doi: 10.12834/VetIt.2117.11956.1

  1. Conclusion: the authors need to highlight the food safety and One Health implication of their findings in the conclusion.

Author Response

Please see the attachment responses to Reviewer 2 comments

Round 2

Reviewer 2 Report

The authors have slightly improved the manuscript but did not satisfactorily address critical issues raised especially in comments numbers 4, 5, 11 and 13.

My previous comments on these still apply and need to be addressed.

Comment 5: 0.15% to 0.41%? How about this: Global Trends in Antimicrobial Use in Food Animals from 2017 to 2030. Antibiotics, doi: 10.3390/antibiotics9120918.?

Comment 11: Which sampling method/technique - probability or non probability? Mention the exact type used.

Sample size calculation: Why 7% error margin and 93% CL? Please don't exceed 5% error margin and use 95% CL. If you correctly compute the sample size using 63% prevalence as claimed, you will get more than 400 poultry farms to survey. I guess your computation was wrong.

Again, the figure legends and Table titles are cannot stand alone. readers should be able to read the titles/legends and understand everything about the figure or the table without referring  to the body of the paper.

For figure 1 legend: "...used..."

Newly introduced references were not edited according to the journal's style

Author Response

Please see the attachment, responses to Reviewer 2 comment, round 2.
